# INTELLIGIBLE LANGUAGE MODELING WITH INPUT SWITCHED AFFINE NETWORKS

**Jakob N. Foerster,**[*] **Justin Gilmer,**[†] **Jan Chorowski,**[‡] **Jascha Sohl-Dickstein, David Sussillo**
Google Brain
Google Inc.
Mountain View, CA 94043, USA
`jakob.foerster@cs.ox.ac.uk, jan.chorowski@cs.uni.wroc.pl`
`{gilmer, jaschasd, sussillo}@google.com`

## ABSTRACT

The computational mechanisms by which nonlinear recurrent neural networks (RNNs) achieve their goals remains an open question. There exist many problem domains where intelligibility of the network model is crucial for deployment. Here we introduce a recurrent architecture composed of input-switched affine transformations, in other words an RNN without any nonlinearity and with one set of weights per input. We show that this architecture achieves near identical performance to traditional architectures on language modeling of Wikipedia text, for the same number of model parameters. It can obtain this performance with the potential for computational speedup compared to existing methods, by precomputing the composed affine transformations corresponding to longer input sequences. As our architecture is affine, we are able to understand the mechanisms by which it functions using linear methods. For example, we show how the network linearly combines contributions from the past to make predictions at the current time step. We show how representations for words can be combined in order to understand how context is transferred across word boundaries. Finally, we demonstrate how the system can be executed and analyzed in arbitrary bases to aid understanding.

## 1 INTRODUCTION

Neural networks and the general field of deep learning have made remarkable progress over the last few years in fields such as object recognition (Krizhevsky et al., 2012), language translation (Sutskever et al., 2014), and speech recognition (Graves et al., 2013). For all of the success of the deep learning approach however, there are certain application domains in which intelligibility of the system is an essential design requirement. One commonly used example is the necessity to understand the decisions that a self-driving vehicle makes when avoiding various obstacles in its path. Another example is the application of neural network methodologies to scientific discovery (Mante et al., 2013). Even where intelligibility is not an overt design requirement, it is fair to say that most users of neural networks would like to better understand the models they deploy.

There are at least two approaches to creating intelligible network models. One approach is to build networks as normal, and then apply analysis techniques after training. Often this approach yields systems that perform extremely well, and whose intelligibility is limited. A second approach is to build a neural network where intelligibility is an explicit design constraint. In this case, the typical result is a system that can be understood reasonably well, but may underperform. In this work we follow this second approach and build intelligibility into our network model, yet without sacrificing performance for the task we studied.

Designing intelligibility into neural networks for all application domains is a worthy, but daunting goal. Here we contribute to that larger goal by focusing on a commonly studied task, that of character based

---

[*] This work was performed as an intern at Google Brain.
[†] Work done as a member of the Google Brain Residency program (`g.co/brainresidency`)
[‡] Work performed when author was a visiting faculty at Google Brain.

language modeling. We develop and analyze a model trained on a one-step-ahead prediction task of the Text8 dataset, which is 10 million characters of Wikipedia text (Mahoney, 2011). The model we use is a switched affine system, where the input determines the switching behavior by selecting a transition matrix and bias as a function of that input, and there is no nonlinearity. Surprisingly, we find that this simple architecture performs as well as a vanilla RNN, Gated Recurrent Unit (GRU) (Cho et al., 2014), IRNN (Le et al., 2015), or Long Short Term Memory (LSTM) (Hochreiter & Schmidhuber, 1997) in this task, despite being a simpler and potentially far more computationally efficient architecture.

In what follows, we discuss related work, define our Input Switched Affine Network (ISAN) model, demonstrate its performance on the one-step-ahead prediction task, and then analyze the model in a multitude of ways, most of which would be currently difficult or impossible to accomplish with modern nonlinear recurrent architectures.

## 2 RELATED WORK

Work by the authors of (Karpathy et al., 2015) attempted to use character-based language modeling to begin to understand how the LSTM (Hochreiter & Schmidhuber, 1997) functions. In it, they employ n-gram word models to highlight what the LSTM has – and has not – learned about the text corpus. They were able to break down LSTM language model errors into classes, such as e.g., "rare word" errors. The authors of (Greff et al., 2015) engaged in a large study to understand the relative importance of the various components of an LSTM. The authors of (Collins et al., 2016) performed an enormous hyperparameter study to disentangle the effects of capacity and trainability in a number of RNN architectures.

Attempts to understand networks in more general contexts include the use of linearization and nonlinear dynamical systems theory to understand RNNs in (Sussillo & Barak, 2013). In feed-forward networks the use of linear probes has been suggested by (Alain & Bengio, 2016), and there exist a host of back-propagation techniques used to infer the most important input to drive various components of the feed-forward network, e.g. (Le et al., 2012).

The ISAN uses an input-switched affine model. The highly related linear time-varying systems are standard material in undergraduate electrical engineering text books. Probabilistic versions of switching linear models with discrete latent variables have a history in the context of probabilistic graphical models. A recent example is the switched linear dynamical system in (Linderman et al., 2016). Focusing on language modeling, (Belanger & Kakade, 2015) defined a probabilistic linear dynamical system as a generative language model for creating context-dependent token embeddings and then used steady-state Kalman filtering for inference over token sequences. They used singular value decomposition and discovered that the right and left singular vectors were semantically and syntactically related. One difference between the ISAN and the LDS is that the weight matrices of the ISAN are input token dependent (while the biases of both models are input dependent). Finally, multiplicative neural networks (MRNNs) were proposed precisely for character based language modeling in (Sutskever et al., 2011; Martens & Sutskever, 2011). The MRNN architecture is similar to our own, in that the dynamics matrix switches as a function of the input character. However, the MRNN relied on a $\tanh$ nonlinearity, while our model is explicitly linear. It is this property of our model which makes it both amenable to analysis, and computationally efficient.

The Observable Operator Model (OOM) (Jaeger, 2000) is similar to the ISAN in that the OOM updates a latent state using a separate transition matrix for each input symbol and performs probabilistic sequence modeling. Unlike the ISAN, the OOM requires that a linear projection of the hidden state corresponds to a normalized sequence probability. This imposes strong constraints on both the model parameters and the model dynamics, and restricts the choice of training algorithms. In contrast, the ISAN applies an affine readout to the hidden state to obtain logits, which are then pushed through a SoftMax to obtain probabilities. Therefore no constraints need to be imposed on the ISAN's parameters and training is easy using backprop. Lastly, the ISAN is formulated as an affine, rather than linear model. While this doesn't change the class of processes that can be modeled, it enhances the stability of training and greatly enhances interpretability. We elaborate upon these ideas in Section 6.1.

## 3 METHODS

### 3.1 MODEL DEFINITION

In what follows $\mathbf{W_x}$ and $\mathbf{b_x}$ respectively denote a transition matrix and a bias vector for a specific input $\mathbf{x}$, the symbol $\mathbf{x}_t$ is the input at time $t$, and $\mathbf{h}_t$ is the hidden state at time $t$. Our ISAN model is defined as

$$\mathbf{h}_t = \mathbf{W_{x_t}}\, \mathbf{h}_{t-1} + \mathbf{b_{x_t}}. \tag{1}$$

The network also learns an initial hidden state $\mathbf{h}_0$. We emphasize the intentional absence of any nonlinear activation function.

### 3.2 CHARACTER LEVEL LANGUAGE MODELLING WITH RNNS

The RNNs are trained on the Text8 Wikipedia dataset, for one-step-ahead character prediction. The Text8 dataset consists only of the 27 characters 'a'-'z' and '_' (space). Given a character sequence of $\mathbf{x}_1, ..., \mathbf{x}_t$, the RNNs are trained to minimize the cross-entropy between the true next character, and the output prediction. We map from the hidden state, $\mathbf{h}_t$, into a logit space via an affine map. The probabilities are computed as

$$p\left(\mathbf{x}_{t+1}\right) = \text{softmax}\left(\mathbf{l}_t\right) \tag{2}$$
$$\mathbf{l}_t = \mathbf{W}_{ro}\, \mathbf{h}_t + \mathbf{b}_{ro}, \tag{3}$$

where $\mathbf{W}_{ro}$ and $\mathbf{b}_{ro}$ are the readout weights and biases, and $\mathbf{l}_t$ is the logit vector. In line with (Collins et al., 2016) we split the training data into 80%, 10%, and 10% for train, test, and evaluation set respectively. The network was trained with the same hyperparameter tuning infrastructure as in (Collins et al., 2016). Analysis in this paper is carried out on the best-performing ISAN model, which has $1,271,619$ parameters, corresponding to 216 hidden units, and 27 dynamics matrices $\mathbf{W_x}$ and biases $\mathbf{b_x}$.

## 4 RESULTS AND ANALYSIS

### 4.1 ISAN PERFORMANCE ON THE TEXT8 TASK

The results on Text8 are shown in Figure 1a. For the largest parameter count, the ISAN matches almost exactly the performance of all other nonlinear models with the same number of maximum parameters: RNN, IRNN, GRU, LSTM. However, we note that for small numbers of parameters the ISAN performs considerably worse than other architectures. All analyses use ISAN trained with 1.28e6 maximum parameters (1.58 bpc cross entropy). Samples of generated text from this model are relatively coherent. We show two examples, after priming with "annual reve", at inverse temperature of 1.5, and 2.0, respectively:

- *"annual revenue and producer of the telecommunications and former communist action and saving its new state house of replicas and many practical persons"*

- *"annual revenue seven five three million one nine nine eight the rest of the country in the united states and south africa new"*.

As a preliminary, comparative analysis, we performed PCA on the state sequence over a large set of sequences for the vanilla RNN, GRU of varying sizes, and ISAN. This is shown in Figure 1b. The eigenvalue spectra, in log of variance explained, was significantly flatter for the ISAN than the other architectures.

We also compare the ISAN performance to a fully linear RNN without input switched dynamics. This achieves a cross-entropy of 3.1 bits / char, independent of network size. This perplexity is only slightly better than that of a Naive Bayes model on the task, at 3.3 bits / char. The output probability of the fully linear network is a product of contributions from each previous character, as in Naive Bayes. Those factorial contributions are learned however, giving ISAN a slight advantage. We also run a comparison to a fully linear network with a non-linear readout. This achieves 2.15 bits /

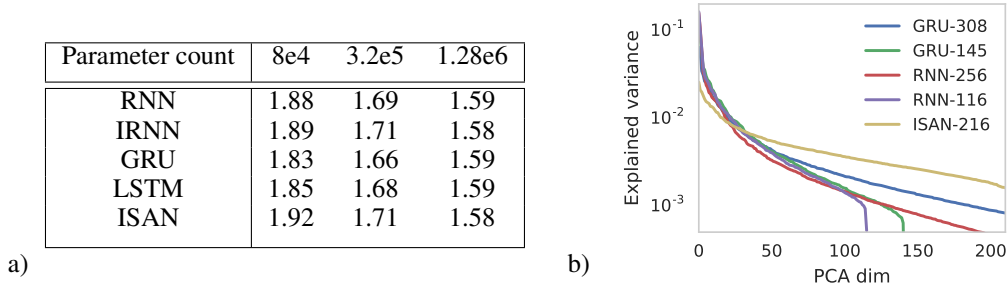

| Parameter count | 8e4 | 3.2e5 | 1.28e6 |
|---|---|---|---|
| RNN | 1.88 | 1.69 | 1.59 |
| IRNN | 1.89 | 1.71 | 1.58 |
| GRU | 1.83 | 1.66 | 1.59 |
| LSTM | 1.85 | 1.68 | 1.59 |
| ISAN | 1.92 | 1.71 | 1.58 |

a)    b)

Figure 1: The ISAN has near identical performance to other RNN architectures, and makes fuller use of its latent space. *a)* Performance of RNN architectures on Text8 one-step-ahead prediction, measured as cross-entropy loss on a held-out test set, in bits per character. The loss is shown as a function of the maximum number of parameters a model is allowed. The values reported for all other architectures are taken from (Collins et al., 2016). *b)* The explained variance ratio of the first 210 most significant PCA dimensions of the hidden states across several architectures. The legend provides the number of latent units for each architecture. We find the ISAN model uses the hidden space more uniformly than the vanilla RNN or GRU.

char, independent of network size. Both of these comparisons illustrate the importance of the input switched dynamics for achieving good results in the absence of non-linear hidden state dynamics.

Lastly we also test to what extent the ISAN can deal with large dictionaries by running it on a byte-pair encoding of the text8 task, where the input dictionary consists of the $27^2$ different possible character combinations. We find that in this setup the LSTM consistently outperforms the ISAN for the same number of parameters. At $1.3m$ parameters the LSTM achieves a cross entropy of 3.4 bits / char-pair, while ISAN achieves 3.55. One explanation for this finding is that the matrices in ISAN are a factor of 27 smaller than the matrices of the LSTMs. For very large numbers of parameters the performance of any architecture saturates in the number of parameters, at which point the ISAN can 'catch-up' with more parameter efficient architectures like LSTMs.

## 4.2 DECOMPOSITION OF CURRENT PREDICTIONS BASED ON PREVIOUS TIME STEPS

Taking advantage of the linearity of the hidden state dynamics for any sequence of inputs, we can decompose the current latent state $\mathbf{h}_t$ into contributions originating from different timepoints $s$ in the history of the input:

$$\mathbf{h}_t = \sum_{s=0}^{t} \left( \prod_{s'=s+1}^{t} \mathbf{W}_{\mathbf{x}_{s'}} \right) \mathbf{b}_{\mathbf{x}_s}, \tag{4}$$

where the empty product when $s + 1 > t$ is 1 by convention, and $\mathbf{b}_{\mathbf{x}_0} = \mathbf{h}_0$ is the learned initial hidden state. This is useful because we can analyze which factors were important in the past, for determining the current character prediction.

Using this decomposition and the linearity of matrix multiplication we can also write the unnormalized logit-vector, $\mathbf{l}_t$, as a sum of terms linear in the biases,

$$\mathbf{l}_t = \mathbf{b}_{ro} + \sum_{s=0}^{t} \boldsymbol{\kappa}_s^t \tag{5}$$

$$\boldsymbol{\kappa}_s^t = \mathbf{W}_{ro} \left( \prod_{s'=s+1}^{t} \mathbf{W}_{\mathbf{x}_{s'}} \right) \mathbf{b}_{\mathbf{x}_s}, \tag{6}$$

where $\boldsymbol{\kappa}_s^t$ is the contribution from timestep $s$ to the logits at timestep $t$, and $\boldsymbol{\kappa}_t^t = \mathbf{b}_{\mathbf{x}_t}$. For notational convenience we will sometimes replace the subscript $s$ with the corresponding input character $\mathbf{x}_s$ at step $s$ when referring to $\boldsymbol{\kappa}_s^t$ – e.g. $\boldsymbol{\kappa}_{\text{'q'}}^t$ to refer to the contribution from the character 'q' in a string.

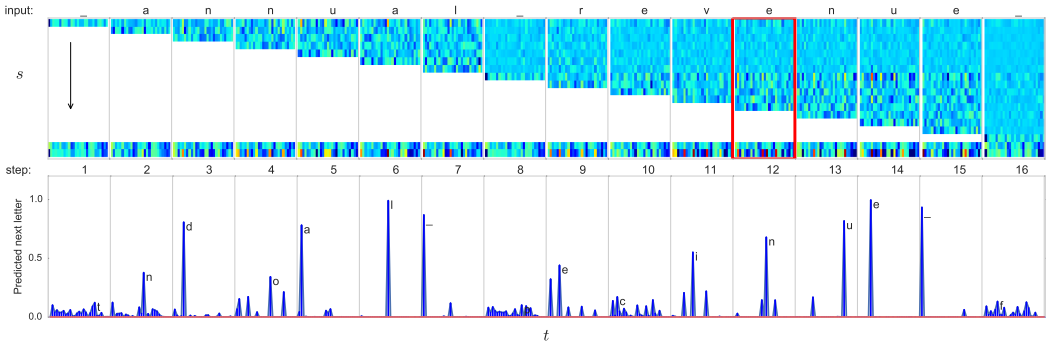

Figure 2: Using the linearity of the hidden state dynamics, predictions at step $t$ can be broken out into contributions, $\kappa_s^t$, from previous steps. Accordingly, each row of the top pane corresponds to the propagated contribution ($\kappa_s^t$) of the input character at time $s$, to the prediction at time $t$ (summed to create the logit at time $t$). The penultimate row contains the output bias vector replicated at every time step. The last row contains the logits of the predicted next character, which is the sum of all rows above. The bottom pane contains the corresponding softmax probabilities at each time $t$ for all characters (time is separated by gray lines). Labeled is the character with the maximum predicted probability. The timestep boxed in red is examined in more detail in Figure 3.

Similarly, when discussing the summed contributions from a word or substring we will sometimes write $\kappa_{word}^t$ to mean the summed contributions of all the $\kappa_s^t$ from that source word, $\sum_{s \in \text{word}} \kappa_s^t$ — e.g. $\kappa_{\text{'the'}}^t$ to refer to the total logit contribution from the word 'the'.

While in standard RNNs the nonlinearity causes interdependence of the bias terms across time steps, in the ISAN the bias terms can be interpreted as independent linear contributions to the state that are propagated and transformed through time. We emphasize that $\kappa_s^t$ includes the multiplicative contributions from the $\mathbf{W}_{\mathbf{x}_{s'}}$ for $s < s' \leq t$. It is however independent of prior inputs, $\mathbf{x}_{s'}$ for $s' < s$. This is the main difference between the analysis we can carry out with the ISAN compared to a non-linear RNN. In general the contribution of a specific character sequence will depend on the hidden state at the start of the sequence. Due to the linearity of the dynamics, this dependency does not exist in the ISAN.

In Figure 2 we show an example of how this decomposition allows us to understand why a particular prediction is made at a given point in time, and how previous characters influence the decoding. For example, the sequence '_annual_revenue_' is processed by the ISAN: Starting with an all-zero hidden state, we use equation (6) to accumulate a sequence of $\kappa_{\text{'_'}}^t, \kappa_{\text{'a'}}^t, \kappa_{\text{'n'}}^t, \kappa_{\text{'n'}}^t, \ldots$. These values can then be used to understand the prediction of the network at some time $t$, by simple addition across the $s$ index, which is shown in Figure 2.

In Figure 3 we provide a detailed view of how past characters contribute to the logits predicting the next character. There are two competing options for the next letter in the word stem 'reve': either 'revenue' or 'reverse'. We show that without the contributions from '_annual' the most likely decoding of the character after the second 'e' is 'r' (to form 'reverse'), while the contributions from '_annual' tip the balance in favor of 'n', decoding to 'revenue'. In a standard RNN a similar analysis could be carried out by comparing the prediction given an artificially limited history.

Using the decomposition of current step predictions in to $\kappa_s^t$, we can also investigate how quickly the contributions of $\kappa_s^t$ decay as a function of $t - s$. In Figure 4a we can see that this contribution decays on two different exponential timescales. We hypothesize that the first time scale corresponds to the decay within a word, while the next corresponds to the decay of information across words and sentences. This effect is also visible in Figure 5. We note that it would be difficult to carry out this analysis in a non-linear RNN.

We can also show the relevance of the $\kappa_s^t$ contributions to the decoding of characters at different positions in the word. For examples, we observe that $\kappa_{\text{'_'}}^t$ makes important contributions to the prediction of the next character at time $t$. We show that using only the $\kappa_{\text{'_'}}^t$, the model can achieve

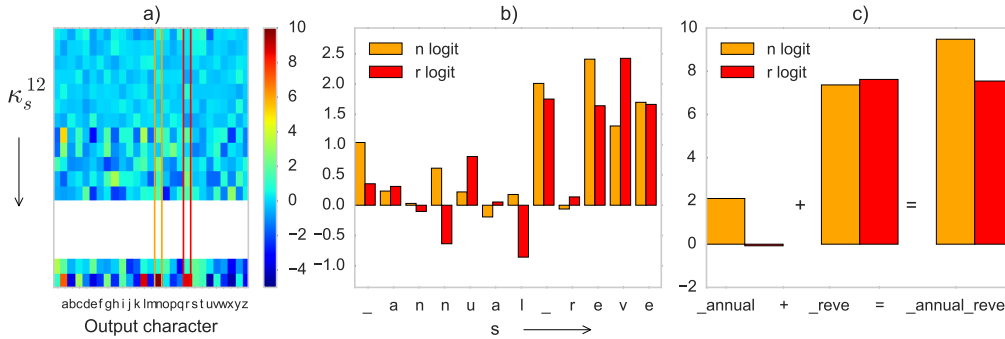

Figure 3: Detailed view of the prediction stack for the final 'n' in '_annual_revenue'. In *a)* all $\kappa_s^t$ are shown, in *b)* only the contributions to the 'n' logit and 'r' logits are shown, in orange and red respectively, from each earlier character in the string. This corresponds to a zoom in view of the columns highlighted in orange and red in a). In *c)* we show how the sum of the contributions from the string '_annual', $\kappa_{:\text{annual}}^t$, pushes the prediction at '_annual_reve' from 'r' to 'n'. Without this contribution the model decodes based only on $\kappa_{:\text{reve}}^t$, leading to a MAP prediction of 'reverse'. With the contribution from $\kappa_{:\text{annual}}^t$ it instead predicts 'revenue'. The contribution of $\kappa_{:\text{annual}}^t$ to the 'n' and 'r' logits is purely linear.

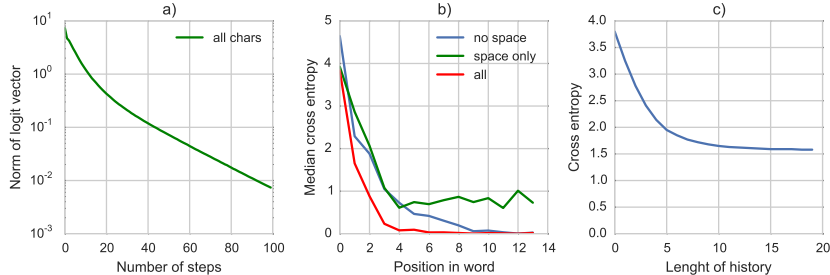

Figure 4: The time decay of the contributions from each character to prediction. *a)* Average norm of $\kappa_s^t$ across training text, $\mathbb{E}\left[||\kappa_s^t||_2\right]$, plotted as a function of $t - s$, and averaged across all source characters. The norm appears to decay exponentially at two rates, a faster rate for the first ten or so characters, and then a slower rate for more long term contributions. *b)* The median cross entropy as a function of the position in the word under three different circumstances: the red line uses all of the $\kappa_s^t$ (baseline), the green line sets all $\kappa_s^t$ apart from $\kappa_{:\_}^t$ to zero, while the blue line only sets $\kappa_{:\_}^t$ to zero. The results from panel c demonstrate the disproportionately large importance of '_' in decoding, especially at the onset of a word. *c)* Shown is the cross-entropy as a function of history when artificially limiting the number of characters available for prediction. This corresponds to only considering the most recent $n$ of the $\kappa$, where $n$ is the length of the history.

a cross entropy of $< 1$ / char when the position of the character is more than 3 letters from the beginning of the word.

Furthermore we can link back from the norm-decay to the importance of past characters for the decoding quality. By artificially limiting the number of past $\kappa$ available for prediction, Figure 4c, we show that the prediction quality improves rapidly when extending the history from 0 to 10 characters and then saturates. This rapid improvement aligns with the range of faster decay in Figure 4a.

## 4.3 FROM CHARACTERS TO WORDS

The ISAN provides a natural means of moving from character level representation to word level. Using the linearity of the hidden state dynamics we can aggregate all of the $\kappa_s^t$ belonging to a given

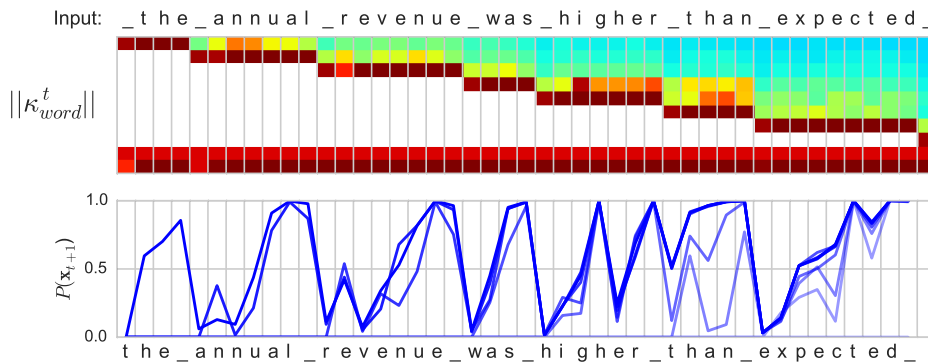

Figure 5: The ISAN architecture can be used to precisely characterize the relationship between words and characters. The top pane shows how exploiting the linearity of the network's operation we can combine the $\kappa_{s_1}^t .. \kappa_{s_n}^t$ in a word to a single contribution, $\kappa_{word}^t$, for each word. Shown is the norm of $\kappa_{word}^t$, a measure of the magnitude of the effect of the previous word on the selection of the current character (red corresponds to a norm of $10$, blue to $0$). The bottom pane shows the probabilities assigned by the network to the next sequence character. Lighter lines show predictions conditioned on a decreasing number of preceding words. For example, when predicting the characters of 'than' there is a large contribution from both $\kappa_{\cdot,\text{was}}^t$ and $\kappa_{\cdot,\text{higher}}^t$, as shown in the top pane. The effect on the log probabilities can be seen in the bottom pane as the model becomes less confident when excluding $\kappa_{\cdot,\text{was}}^t$ and significantly less confident when excluding both $\kappa_{\cdot,\text{was}}^t$ and $\kappa_{\cdot,\text{higher}}^t$. This word based representation clearly shows that the system leverages contextual information across multiple words.

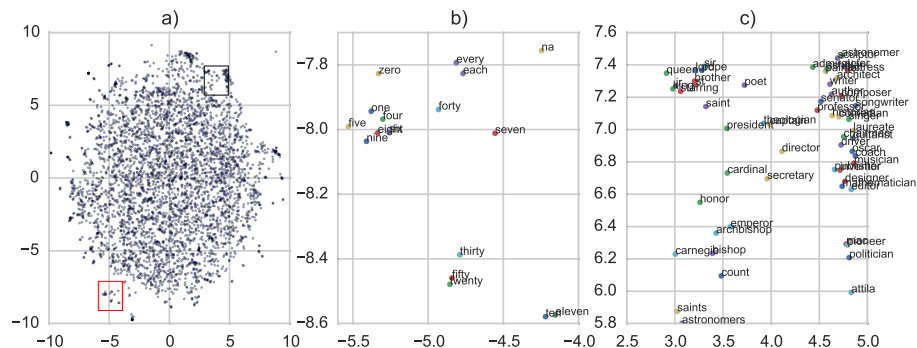

Figure 6: We can use the $\kappa_{word}^t$ as an embedding space. Although the model was only trained on character level representations, the $\kappa_{word}^t$ show clear semantic clustering under t-SNE (Maaten & Hinton, 2008). Shown is an overview of the 4000 most common words in a). In b) a zoomed in version is shown, a region that is primarily filled with numbers. In c) the zoom captures a variety of different professions.

word and visualize them as a single contribution to the prediction of the letters in the next word. This allows us to understand how each preceding word impacts the decoding for the letters of later words. In Figure 5 we show that the words 'higher' and 'than' make large contributions to the prediction of the characters 'h' and 'n' in 't**eve**nue', as measured by the norm of the $\kappa_{\cdot,\text{the}}^t$ and $\kappa_{\cdot,\text{annual}}^t$.

In Figure 6 we show that these $\kappa_{word}^t$ are more than a mathematical convenience and even capture word-level semantic information. Shown is a t-SNE embedding of the $\kappa_{word}^t$ for the most common 4000 words in the data-set, with examples of the kind of clusters that arise.

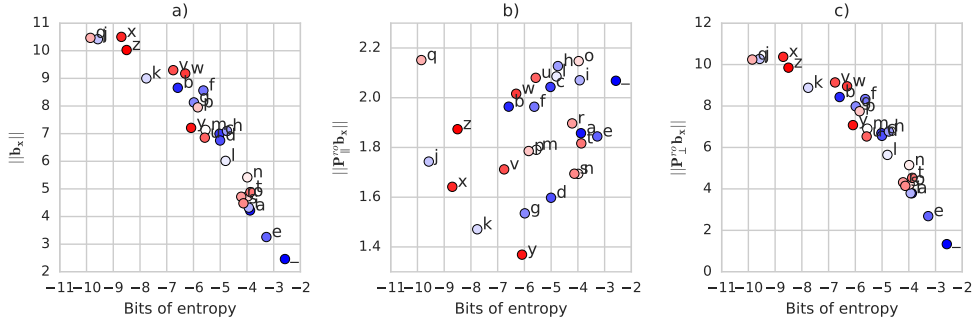

Figure 7: By transforming the ISAN dynamics into a new basis, we can better understand the action of the input-dependent biases. *a)* We observe a strong correlation between the norms of the input dependent biases, $\mathbf{b_x}$, and the log-probability of the unigram $\mathbf{x}$ in the training data. We can begin to understand this correlation structure using a basis transform into the 'readout basis'. Breaking out the norm into its components in $\mathbf{P}_{\parallel}^{ro}$ and $\mathbf{P}_{\perp}^{ro}$ in *b)* and *c)* respectively, shows that the correlation is due to the component orthogonal to $\mathbf{W}_{ro}$. This implies a connection between information or 'surprise' and distance in the 'computational' subspace of state space.

## 4.4 CHANGE OF BASIS

We are free to perform a change of basis on the hidden state, and then to run the affine ISAN dynamics in that new basis. Note that this change of basis is not possible for other RNN architectures, since the action of the nonlinearity depends on the choice of basis.

In particular we can construct a 'readout basis' that explicitly divides the latent space into a subspace $\mathbf{P}_{\parallel}^{ro}$ spanned by the rows of the readout matrix $\mathbf{W}_{ro}$, and its orthogonal complement $\mathbf{P}_{\perp}^{ro}$. This representation explicitly divides the hidden state dynamics into a 27-dimensional 'readout' subspace that is accessed by the readout matrix to make predictions, and a 'computational' subspace comprising the remaining $216 - 27$ dimensions that are orthogonal to the readout matrix.

We apply this change of basis to analyze an intriguing observation about the hidden offsets $\mathbf{b_x}$: As shown in Figure 7, the norm of the $\mathbf{b_x}$ is strongly correlated to the log-probability of the unigram $\mathbf{x}$ in the training data. Re-expressing network parameters using the 'readout basis' shows that this correlation is not related to reading out the next-step prediction. This is because the norm of the projection of $\mathbf{b_x}$ into $\mathbf{P}_{\perp}^{ro}$ remains strongly correlated with character frequency, while the projections into $\mathbf{P}_{\parallel}^{ro}$ have norms that show little correlation. This indicates that the information content or surprise of a letter is encoded through the norm of the component of $\mathbf{b_x}$ in the computational space, rather than in the readout space.

Similarly, in Figure 8 we illustrate that the structure in the correlations between the $\mathbf{b_x}$ is due to their components in $\mathbf{P}_{\parallel}^{ro}$, while the correlation in $\mathbf{P}_{\perp}^{ro}$ is relatively uniform. We can clearly see two blocks of high correlations between the vowels and consonants respectively, while $\mathbf{b}._{\_}$ is uncorrelated to either.

## 4.5 COMPARISON WITH $n$-GRAM MODEL WITH BACKOFF

We compared the computation performed by $n$-gram language models and those performed by the ISAN. An n-gram model with back-off weights expresses the conditional probability $p\left(\mathbf{x}_t|\mathbf{x}_1...\mathbf{x}_{t-1}\right)$ as a sum of smoothed count ratios of $n$-grams of different lengths, with the contribution of shorter $n$-grams down-weighted by backoff weights. On the other hand, the computations performed by the ISAN start with the contribution of $\mathbf{b}_{ro}$ to the logits, which as shown in Figure 9a) corresponds to the unigram log-probabilities. The logits are then additively updated with contributions from longer $n$-grams, represented by $\boldsymbol{\kappa}_s^t$. This additive contribution to the logits corresponds to a multiplicative modification of the emission probabilities from histories of different length. For long time lags, the additive correction to log-probabilities becomes small (Figure 2), which corresponds to multiplication by a uniform distribution. Despite these differences in how n-gram history is incorporated, we

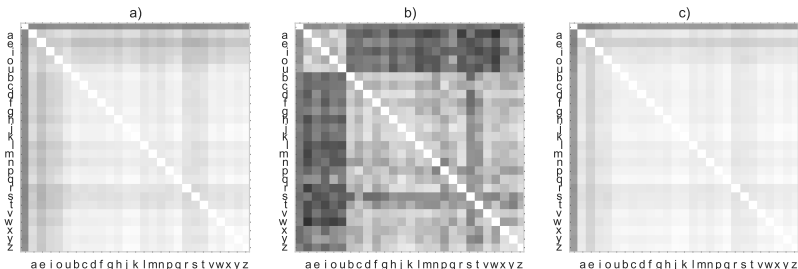

Figure 8: By transforming ISAN dynamics into a new basis, we can better interpret structure in the input-dependent biases. In *a)* we show the cosine distance between the input dependent bias vectors, split between vowels and consonants (' ' is first). In *b)* we show the correlation only considering the components in the subspace $\mathbf{P}_{\parallel}^{ro}$ spanned by the rows of the readout matrix $\mathbf{W}_{ro}$. *c)* shows the correlation of the components in the orthogonal complement $\mathbf{P}_{\perp}^{ro}$. In all plots white corresponds to 0 (aligned) and black to 2.

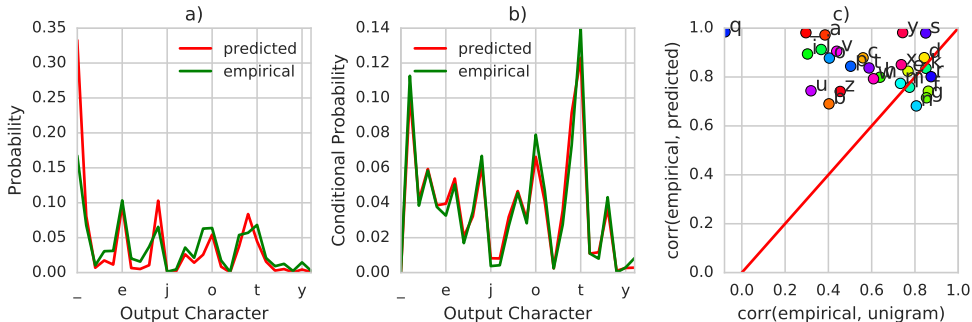

Figure 9: The predictions of ISAN for one and two characters well approximate the predictions of unigram and bigram models. In *a)* we compare softmax($\mathbf{b}_{ro}$) to the empirical unigram distribution $P(\mathbf{x})$. In *b)* we compare softmax($\mathbf{W}_{ro}\mathbf{b}_{\cdot\cdot} + \mathbf{b}_{ro}$) with the empirical distribution $P(\mathbf{x}_{t+1}|\text{'\_'})$. In *c)* we show the correlation of softmax($\mathbf{W}_{ro}\mathbf{b}_{\mathbf{x}} + \mathbf{b}_{ro}$) with $P(\mathbf{x}_{t+1}|\mathbf{x}_t)$ for all 27 characters (y-axis), and compare this to the correlation between the empirical unigram probabilities $P(\mathbf{x})$ to $P(\mathbf{x}_{t+1}|\mathbf{x}_t)$ (x-axis). The plot shows that the readout of the bias vector is a better predictor of the conditional distribution than the unigram probability.

nevertheless observe an agreement between empirical models estimated on the training set and model predictions for unigrams and bigrams. Figure 9 shows that the bias term $\mathbf{b}_{ro}$ gives the unigram probabilities of letters, while the addition of the offset terms $\mathbf{b}_{\mathbf{x}}$ accurately predict the bigram distribution of $P(\mathbf{x}_{t+1}|\mathbf{x}_t)$. Shown are both an example, $P(\mathbf{x}|\text{'\_'})$, and a summary plot for all 27 letters.

We further explore the $n$-gram comparison by artificially limiting the length of the character history that is available to the ISAN for making predictions, as shown in Figure 4c).

## 5 ANALYSES OF PARENTHESES COUNTING TASK

To show the interpretability of the ISAN we train a model on the parenthesis counting task. Bringing together ideas from sections 4.4 and 6.1 we re-express the transition dynamics in a new basis that fully reveals computations performed by the ISAN.

We analyze the simple task of parentheses counting, which was defined in (Collins et al., 2016). Briefly, the RNN is required keep track of the nesting level of 3 different types of parentheses

independently. The inputs are the one-hot encoding of the different opening and closing parentheses (e.g. '(', ')', '{', '}') as well as a noise character ('a'). The output is the one-hot encoding of the nesting level between (0-5), one set of counts for each parentheses task. One change from (Collins et al., 2016) is that we slightly simplify the problem by exchanging the cross-entropy error with an $L2$ error and linear readout (this change leads to slightly cleaner figures, but does not qualitatively change the results).

We first re-express the transition dynamics in terms of linear, rather than affine operations. Consider the matrix $\mathbf{W}' \in \mathcal{R}^{(n+1)\times(n+1)}$:

$$\mathbf{W}' = \begin{bmatrix} \mathbf{W} & \mathbf{b} \\ \mathbf{0} & 1 \end{bmatrix}, \tag{7}$$

where $\mathbf{0}$ is a row vector of zeros. The matrix $\mathbf{W}'$ emulates the affine transition for any hidden state $\mathbf{h} \in \mathcal{R}^{n\times1}$:

$$\mathbf{W}' \begin{bmatrix} \mathbf{h} \\ 1 \end{bmatrix} = \begin{bmatrix} \mathbf{W}\mathbf{h} + \mathbf{b} \\ 1 \end{bmatrix}. \tag{8}$$

The matrices $\mathbf{W}$ and $\mathbf{W}'$ are closely connected. Each eigenvalue of $\mathbf{W}$ is also an eigenvalue of $\mathbf{W}'$. Moreover, eigenvectors of $\mathbf{W}$ become the eigenvectors of $\mathbf{W}'$ when expanded with a zero dimension. In fact, $\mathbf{W}'$ only has one extra eigenvalue of exactly 1 that is necessary to preserve the last dimension of the expanded hidden state.

To analyze the the parentheses task we analyze $\mathbf{W}'$. The key to understanding how the network solves the parentheses task is to find a change of bases that clarifies the dynamics necessary to count 3 sets of independent parentheses nesting levels. In this case we use a matrix composed of the readout matrix, modified by adding a set of vectors that spans the null space of the readout (including the additional bias dimension).

$$\mathbf{W}'_{ro} = \begin{bmatrix} \mathbf{W}_{ro} & \mathbf{b}_{ro} \\ \mathbf{O} \end{bmatrix} \tag{9}$$

where $\mathbf{O}$ is the orthogonal complement of the subspace spanned by the row vectors of $[\mathbf{W}_{ro}\ \mathbf{b}_{ro}]$ in an $N+1$-dimensional space. We perform the following change of basis of the dynamics matrices,

$$\mathbf{W}^{x}_{(ro)}{}' = \mathbf{W}_{ro}{}'\mathbf{W}^{x'}\left(\mathbf{W}'_{ro}\right)^{-1} \tag{10}$$

and visualize the results in Fig. 10. Figure 10 shows that this system created delay lines which count the nesting level, with fixed point dynamics at the 0 count (5 count), so that the system stays at both numbers when the input would otherwise increment (decrement) the count. The matrices also implement fixed point dynamics, as implemented via an identity submatrix to preserve the memory of the parenthesis nesting counts when an unrelated symbol enters the system (e.g. The '{}' count is preserved by the '(' matrix when a '(' symbol enters the system).

# 6 DISCUSSION

In this paper we motivated an input-switched affine recurrent network for the purpose of intelligibility. We showed that a switched affine architecture achieves the same performance, for the same number of maximum parameters, on a language modeling task as do more common RNN architectures, including GRUs and LSTMs. We performed a series of analyses, demonstrating that the simplicity of the latent dynamics makes the trained RNN far easier to understand and interpret.

## 6.1 BENEFITS OF AFFINE TRANSITIONS OVER LINEAR

ISAN uses affine operators to model state transitions assigned to each input symbol. Following eq. (1) each transition consists of matrix multiplication and bias vector addition. An important question is whether the biases are needed and how the ISAN would be impacted if linear transition operators were used instead of affine ones. The answer is two-fold. First, affine dynamics can be exactly implemented using linear operators in a hidden space expanded by one additional dimension. Therefore, the expressivity of ISAN does not depend on choosing a linear or affine formulation. However, we found that the affine parametrization of transitions is much easier to train. We attempted to train models using only linear transitions, but achieved a loss of only 4.1 bits per character, which

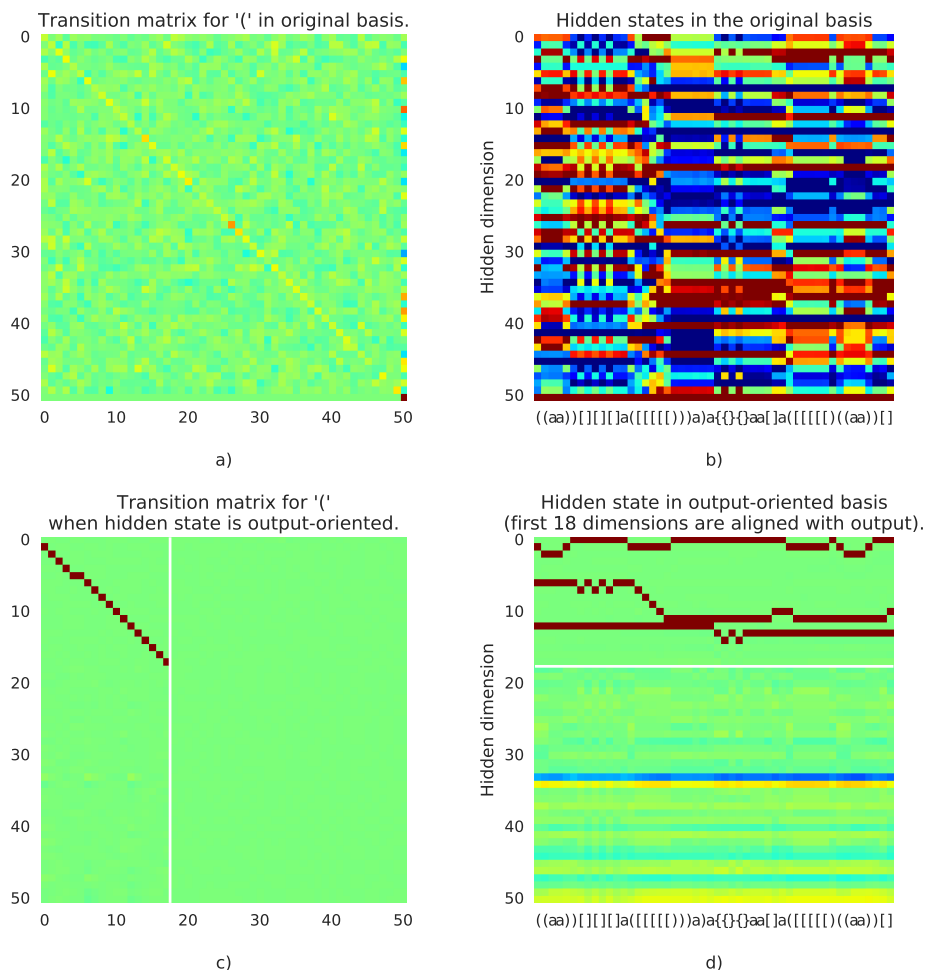

Figure 10: A visualization of the dynamics of an ISAN for the parenthesis counting task. In *a)* the weight matrix for '(' is shown in the original basis. In *c)* it is shown transformed to highlight the delay-line dynamics. The activations of the hidden units are also shown *b)* in the original basis, and *d)* rotated to the same basis as in c), to highlight the delay-line dynamics in a more intelligible way. The white line delineates the transition matrix elements and hidden state dimensions that directly contribute to the output. All matrices for parentheses types appear similarly, with closing parentheses, e.g. ')', changing the direction of the delay line.

corresponds to the performance of a unigram character model. Second, affine operators are easier to interpret because they permit easy visualization of contributions of each input token on the final network's prediction, as demonstrated in Section 4.2.

## 6.2 COMPUTATIONAL BENEFITS

Switched affine networks hold the potential to be massively more computationally and memory efficient for text processing than standard RNNs, as explained in the next two subsections.

### 6.2.1 SPARSE PARAMETER ACCESS

As shown in Figure 1a, the performance for fixed parameter count is nearly identical between the ISAN and other recurrent networks. However, at each time step, only the parameters associated with a single input are used. For $K$ possible inputs and $N$ parameters, the computational cost per update

step is $O\left(\frac{N}{K}\right)$, a factor of $K$ speedup over non-switched architectures. Similarly, the number of hidden units is $O\left(\sqrt{\frac{N}{K}}\right)$, a factor of $K^{\frac{1}{2}}$ memory improvement for storage of the latent state.

### 6.2.2 COMPOSITION OF AFFINE UPDATES

The memory and computational benefits in Section 6.2.1 are shared by other switched networks. However, ISAN is unique in its ability to precompute affine transformations corresponding to input strings. This is possible because the composition of affine transformations is also an affine transformation. This property is used in Section 4.3 to evaluate the linear contributions of words, rather than characters. This means that the hidden state update corresponding to an entire input sequence can be computed with identical cost to the update for a single character (plus the dictionary lookup cost for the composed transformation). ISAN can therefore achieve very large speedups on input processing, at the cost of increased memory use, by accumulating large lookup tables of the $\mathbf{W_x}$ and $\mathbf{b_x}$ corresponding to common input sequences. Of course, practical implementations will have to incorporate complexities of memory management, batching, etc.

### 6.3 FUTURE WORK

There are some obvious future directions to this work. Currently, we define switching behavior using an input set with finite and manageable cardinality. Studying word-level language models with enormous vocabularies may require some additional logic to scale. Adapting this model to continuous-valued inputs is another important direction. One approach is to use a tensor factorization similar to that employed by the MRNN (Sutskever et al., 2014). Another is to build a language model which switches on bigrams or trigrams, rather than characters or words, targeting an intermediate number of affine transformations.

Training very large switched linear models has the potential to be extremely fruitful, due both to their improved computational efficiency, and our ability to better understand and manipulate their behavior.

## 7 ACKNOWLEDGEMENTS

We would like to thank Jasmine Collins for her help and advice, and Quoc Le, David Ha and Mohammad Norouzi for helpful discussions.

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
