# Peer review of "Intelligible Language Modeling with Input Switched Affine Networks"

_ICLR 2017 — rejected_

[Public Comment · Luke Vilnis · 30 Nov 2016]
**Related work**

Very cool work. Here's some possibly related work on treating text as one-hots coming from a latent linear dynamical system over unobserved embeddings, "A Linear Dynamical System Model for Text" by Belanger and Kakade:

[Official Review · AnonReviewer3 · rating 6 · confidence 4 · 17 Dec 2016]
**A character language model that gains some interpretability without large losses in predictivity**

The authors present a character language model that gains some interpretability without large losses in predictivity. 

CONTRIBUTION:

I'd characterize the paper as some experimental investigation of a cute insight.  Recall that multi-class logistic regression allows you to apportion credit for a prediction to the input features: some features raised the probability of the correct class, while others lowered it.  This paper points out that a sufficiently simple RNN model architecture is log-linear in the same way, so you can apportion credit for a prediction among elements of the past history.  

PROS:

The paper is quite well-written and was fun to read.  It's nice to see that a simple architecture still does respectably.
It's easy to imagine using this model for a classroom assignment.  
It should be easy to implement, and the students could replicate the authors' investigation of what influences the network's predictions.
The authors present some nice visualizations.

Section 5.2 also describes some computational benefits.

CAVEATS ON PREDICTIVE ACCURACY:

* Figure 1 says that the ISAN has "near identical performance to other architectures."  But this appears true only when comparing the largest models.  

Explanation: It appears that for smaller parameter sizes, a GRU still beats the authors' model by 22% to 39% in the usual metric of perplexity per word (ppw).  (That's how LM people usually report performance, with a 10% reduction in ppw traditionally being considered a good Ph.D. dissertation.  I assumed an average of 7 chars/word when converting cross-entropy/char to perplexity/word.)  

* In addition, it's not known whether this model family will remain competitive beyond the toy situations tested here.

Explanation: The authors tried it only on character-based language modeling, and only on a 10M-char dataset, so their ppw is extremely high: 2135 for the best models in this paper.  By contrast, a word-based RNN LM trained on 44M words gets ppw of 133, and trained on 800M words gets ppw of 51.  [Numbers copied from the paper I cited before:

[Official Review · AnonReviewer4 · rating 7 · confidence 3 · 19 Dec 2016]
**Original and creative work - hesitation about results.**

Summary: The authors propose an input switched affine network to do character-level language modeling, a kind of RNN without pointwise nonlinearity, but with switching the transition matrix & bias based on the input character. This is motivated by intelligibility, since it allows decomposition of output contribution into these kappa_s^t terms, and use of basic linear algebra to probe the network.

Regarding myself as a reviewer, I am quite sure I understood the main ideas and arguments of this paper, but am not an expert on RNN language models or intelligibility/interpretability in ML.
I did not read any papers with a similar premise - closest related work I'm familiar with would be deconvnet for insight into vision-CNNs.

PRO:
I think this is original and novel work. This work is high quality, well written, and clearly is the result of a lot of work.
I found section 4.5 about projecting into readout subspace vs "computational" subspace most interesting and meaningful.

CON:
+ The main hesitation I have is that the results on both parts (ISAN model, and analysis of it) are not entirely convincing:
   (1) ISAN is only trained on small task (text8), not clear whether it can be a strong char-LM on larger scale tasks,
   (2) nor do the analysis sections provide all that much real insight in the learned network.

(1b) Other caveat towards ISAN architecture: this model in its proposed form is really only fit for small-vocabulary (i.e. character-based) language modeling, not a general RNN with large-vocab discrete input nor continuous input.

(2a) For analysis: many cute plots and fun ideas of quantities to look at, but not much concrete insights.
(2b) Not very clear which analysis is specific to the ISAN model, and which ideas will generalize to general nonlinear RNNs.
(2c) Re sec 4.2 - 4.3: It seems that the quantity \kappa_s^t on which analysis rests, isn't all that meaningful. Elaborating a bit on what I wrote in the question:
For example: Fig 2, for input letter "u" in revenue, there's a red spot where '_' character massively positively impacts the logit of 'e'. This seems quite meaningless, what would be the meaning of influence of '_' character? So it looks ot me that the switching matrix W_u (and prior W_n W_e etc) are using previous state in an interesting way to produce that following e. So that metric \kappa_s^t just doesn't seem very meaningful.
This remark relates to the last paragraph of Sec4.2.

Even though the list of cons here is longer than pro's, I recommend accept; specifically because the originality of this work will in any case make it more vulnerable to critiques. This work is well-motivated, very well-executed, and can inspire many more interesting investigations along these lines.

[Official Review · AnonReviewer2 · rating 6 · confidence 4 · 21 Dec 2016]
**No Title**

Summary:  The authors present a simple RNN with linear dynamics for language modeling. The linear dynamics greatly enhance the interpretability of the model, as well as provide the potential to improve performance by caching the dynamics for common sub-sequences. Overall, the quantitative comparison on a benchmark task is underwhelming. It’s unclear why the authors didn’t consider a more common dataset, and they only considered a single dataset. On the other hand, they present a number of well-executed techniques for analyzing the behavior of the model, many of which would be impossible to do for a non-linear RNN. 

Overall, I recommend that the paper is accepted, despite the results. It provides an interesting read and an important contribution to the research dialogue. 

Feedback

The paper could be improved by shortening the number of analysis experiments and increasing the discussion of related sequence models. Some of the experiments were very compelling, whereas some of them (eg. 4.6) sort of feels like you’re just showing the reader that the model fits the data well, not that the model has any particularly important property. We trust that the model fits the data well, since you get reasonable perplexity results. 

LSTMS/GRUs are great for for language modeling for data with rigid combinatorial structure, such as nested parenthesis. It would have been nice if you compared your model to non-linear methods on this sort of data. Don’t be scared of negative results! It would be interesting if the non-linear methods were substantially better on these tasks. 

You should definitely add a discussion of Belanger and Kakade 2015 to the related work. They have different motivations (fast, scalable learning algorithms) rather than you (interpretable latent state dynamics and simple credit assignment for future predictions given past). On the other hand, they also have linear dynamics, and look at the singular vectors of the transition matrix to analyze the model. 

More broadly, it would be useful for readers if you discussed LDS more directly. A lot of this comparison came up in the openreview discussion, and I recommend folding this into the paper. For example, it would be useful to emphasize that the bias vectors correspond to columns of the Kalman gain matrix. 

One last thing regarding LDS: your model corresponds to Kalman filtering but in an LDS you can also do Kalman smoothing, where state vectors are inferred using the future in addition to the past observations. Could you do something similar in your model?

What if you said that each matrix is a sparse/convex combination of a set of dictionary matrices? This parameter sharing could provide even more interpretability, since the characters are then represented by the low-dimensional weights used to combine the dictionary elements. This could also provide more scalability to word-level problems.

[Public Comment · Jakob Nicolaus Foerster · 16 Jan 2017]
**Overall response to reviews, high level summary of new results and analysis**

We would like to thank all reviewers for their careful and thorough reviews, and for recommending paper acceptance. Based on your specific actionable feedback, we have since improved the paper with additional analysis and experiments. We hope that based on these new results, you will raise your scores and more confidently recommend acceptance.

In particular we have included a new analysis in Section 5 which provides an end-to-end interpretation of the functioning of the ISAN on a parenthesis counting task. For this example we believe we have really `cracked the case’, and fully explain the neural network behavior. We have also included a new analysis (Figure 4c) that quantifies the importance of past characters for current predictions. Furthermore, we have prepared a standalone IPython demo featuring our implementation of the ISAN on the parenthesis counting task and are waiting for approval to release it.
Lastly we have a included a new plot (Figure 6) that uses the \kappa_word as an embedding space, clearly showing that semantic structure arises on a word level, even though the model is only trained on next character prediction. 

In terms of new experimental validation we have added the following comparisons (see Section 4.1):
1) fully linear dynamics (without switching) with linear readouts
2) fully linear dynamics (without switching) but with non-linear readouts 
3) naive bayes
These experiments highlight the crucial importance of input switching for the performance of the ISAN. 

To investigate the limits of the input switched architecture we also ran experiments to compare the ISAN to the LSTM on a word-fragment task with a large number of inputs and find that it performs less well than the LSTM, with a gap of around 0.15 bits / char-pair (details given below).

Finally, we have improved the text in a number of places to address reviewer concerns. These changes are detailed in the per-reviewer responses.

[Public Comment · Jan K Chorowski · 18 Jan 2017]
**Parenthesis counting demo**

We have released the demo of a fully-understandable ISAN network for counting parenthesis at

[Final Decision · Program Chairs · 06 Feb 2017]
**ICLR committee final decision**

All reviewers have carefully looked at the paper and weakly support acceptance of the paper. Program Chairs also looked at this paper and believe that its contribution is too marginal and incremental in its current form. We encourage the authors to resubmit.